# A qualitative study of clinicians' perspectives on independent rights advice for involuntary psychiatric patients in British Columbia, Canada

Iva W. Cheung[1]*, Diego S. Silva[2], Kimberly J. Miller[1,3], Erin E. Michalak[4], Charles H. Goldsmith[1]

1 Faculty of Health Sciences, Simon Fraser University, Burnaby, BC, Canada, 2 Sydney Health Ethics, Sydney School of Public Health, Sydney, NSW, Australia, 3 BC Children's Hospital Research Institute, Vancouver, BC, Canada, 4 Department of Psychiatry, University of British Columbia, Vancouver, BC, Canada

* ivac@sfu.ca, iva.cheung@gmail.com

## Abstract

### Background

In British Columbia (BC), Canada, clinicians are responsible for giving involuntary psychiatric patients rights information upon admission. Yet an investigation by the BC Office of the Ombudsperson found that clinicians are not always fulfilling this responsibility. The Ombudsperson recommended that the provincial government fund an independent body to give rights advice to patients.

### Methods

To understand how clinicians feel about this recommendation, focus groups of clinicians who may give psychiatric patients rights information ($n = 81$) were conducted in Vancouver, BC, to probe their attitudes toward independent rights advisors. The focus group transcripts were thematically analyzed.

### Results

Most clinicians believe that giving rights information is within their scope of practice, although some acknowledge that it poses a conflict of interest when the patient wishes to challenge the treatment team's decisions. Participants' chief concerns about an independent rights-advice service were that (a) patients may experience a delay in receiving their rights information, (b) integrating rights advisors into the workflow would complicate an already chaotic admission process, and (c) more patients would be counselled to challenge their hospitalization, leading to an increased administrative workload for clinical staff. However, many participants believed that independent rights advisors would be a positive addition to the admission process, both allowing clinicians to focus on treatment and serving as a source of rights-related information.

**Data Availability Statement:** The dataset of anonymized focus group transcripts is available via Simon Fraser University's data repository at https://researchdata.sfu.ca/pydio_public/43fd77.

**Funding:** This work was supported by the Canadian Institutes of Health Research (https://cihr-irsc.gc.ca/) under a Doctoral Award to IC (FRN 134903); and the Michael Smith Foundation for Health Research (https://www.msfhr.org/) under a Reach Award to KM, IC, and EM (Grant ID: 17369). The funders had no role in study design; the collection, analysis, or interpretation of data; the writing of the manuscript; or in the decision to submit the article for publication.

**Competing interests:** The authors have declared that no competing interests exist.

## Conclusions

Participants were generally amenable to an independent rights-advice service, suggesting that the introduction of rights advisors need not result in an adversarial relationship between treatment team and patient, as opponents of the proposal fear. Clearly distinguishing between basic rights information and in-depth rights advice could address several of the clinicians' concerns about the role that independent rights advisors would play in the involuntary admission process. Clinicians' and other stakeholders' concerns should be considered as the province develops its rights-advice service.

## Introduction

The *Mental Health Act* in British Columbia (BC), Canada, is the provincial law that sets out the conditions under which people with a mental disorder can be detained as involuntary patients in hospital and receive psychiatric treatment without their consent. According to this law, a person may be involuntarily admitted and treated only if a physician has examined them and believes they meet all four of these criteria:

1. the person has a mental disorder that seriously impairs their ability

    a. to react appropriately to their environment, or

    b. to associate with others;

2. the person requires treatment in or through a designated facility;

3. the person requires care, supervision and control in or through a designated facility to prevent their substantial mental or physical deterioration or for the protection of the person or patient or the protection of others; and

4. the person cannot suitably be admitted as a voluntary patient [1, p. 8].

"In or through a designated facility" refers to the fact that the person may be an involuntary inpatient ("in") at a hospital designated by the Ministry of Health to provide involuntary care or an involuntary outpatient ("through") who receives involuntary treatment but lives in the community.

The physician must complete a medical certificate (also known as the Form 4) to authorize the person's detention. One medical certificate allows the person to be detained for 48 hours. A second certificate completed within that 48-hour period by another physician allows the person to be detained for up to one month from the date of admission. The law does not require either physician completing these certificates to be a psychiatrist. In BC, the law does not require patients to undergo a test for competency before involuntary hospitalization and treatment.

In the 2016–2017 fiscal year, based on discharge records in BC there were 20,483 involuntary hospitalizations, involving 14,980 unique patients. In that same period, the province had 17,656 voluntary hospitalizations involving 11,683 unique patients [2].

Involuntary patients in BC cannot leave the hospital without their physician's permission, and they cannot refuse psychiatric treatment. The ability of the province to restrict a person's freedom of movement and limit their right to informed consent to treatment is an "extraordinary power" [2] that demands safeguards to protect involuntary patients' safety, autonomy, and dignity. Among these safeguards is a patient's right to be informed about the nature of—and possible avenues to challenge—their detention.

According to Section 10 of the *Canadian Charter of Rights and Freedoms* [3], people involuntarily hospitalized under provincial mental health legislation have the right:

- to know why they are being detained,

- to contact a lawyer and to be informed of that right, and

- to challenge the validity of their detention in court (i.e., by way of *habeas corpus*).

Mental health legislation in each province may also specify other rights available to involuntary patients. In BC, for example, patients certified under the *Mental Health Act* have the right to submit a request to the Mental Health Review Board for a review panel hearing to decide whether they meet the criteria for involuntary hospitalization. They also have the right to request a second medical opinion about their treatment plan [4].

Section 34 of the *Mental Health Act* states that involuntary patients must be informed of their rights:

- when they are involuntarily admitted,

- whenever they are transferred to a designated facility, and

- whenever their certification is renewed.

In BC, the task of giving patients rights information is usually assigned to clinical staff, including nurses, social workers, and physicians [5], who are expected to use a statutory document known as Form 13 to convey the information to patients [6]. If, at the time of admission, a patient seems unable to understand the information, the law requires the staff to try offering rights information again later, when the patient appears ready to receive it.

Giving involuntary patients complete information about their rights is not only a legal requirement and an ethical practice to support patients' autonomy [7, article 14], but it can also contribute to giving patients a sense of procedural justice, which can have the therapeutic effects of reducing their feelings of coercion and engaging them in their own care and recovery planning [8,9].

However, a survey commissioned by the BC Ministry of Health and published in 2011 found that 43% of involuntary patients reported not being told their *Mental Health Act* rights in a way they could understand [10]. A 2019 report on an investigation by the BC Office of the Ombudsperson revealed that Form 13 appeared in only 49% of involuntary patients' files, suggesting that clinicians were not fulfilling their obligations under the legislation to give 100% of patients information about their rights and to document this communication [2]. The BC Office of the Ombudsperson recommended in its report that the provincial government establish an independent service to give involuntary patients "timely, in-person" rights advice and support in exercising their rights "based on their particular circumstances" [2, p. 8]. The proposed service would be notified of every involuntary admission, transfer, or renewal, and a rights advisor would visit the patient to "provide information about the patient's status and options as an involuntary patient, advise about the best option given the patient's circumstances, and make referrals to legal counsel where appropriate or where requested by the patient" [2, p. 88]. In other words, this service would ensure that all patients would be informed of their *Mental Health Act* rights in a way they could understand.

This paper presents a study seeking to answer the following question: "What are the attitudes of clinicians who currently give patients rights information in BC toward a proposed proactive, independent rights advice service?" Patients' perspectives of the rights-information process have been presented elsewhere [11]. Understanding stakeholders' concerns about independent rights advice may help the province develop policies that both support

involuntary patients' rights and accommodate the systems clinicians have developed to provide patient care.

## Background

### Arguments against independent rights advice in BC

Critics of the proposal to establish an independent rights-advice service, primarily family caregivers of people with severe mental illness, believe that an independent service would divert resources from mental health care. They argue that having clinicians give patients rights information is no more a conflict of interest than requiring police officers to inform people of their rights when they are arrested. These critics fear that a heavy focus on rights for people who lack insight into their mental illness would lead to premature discharges and deprive them of the treatment they need. They are concerned that rights advisors with anti-psychiatric views may even actively persuade patients to reject treatment and challenge their hospitalization [12].

Further, opponents of independent rights advisors say that clinicians are always available and can give patients rights information immediately upon admission, whereas patients may have to wait for an independent rights advisor, who may not work outside of usual business hours, to become available. These critics also note that independent rights advisors, whom they assume will have legal or paralegal backgrounds, are not trained to interact with people with mental illness in what should be a therapeutic setting, and they fear that independent rights advisors will transform a clinical interaction into an adversarial one [13, pp. 336–352].

Currently, clinicians are merely responsible for giving involuntary patients rights *notification* or rights *information*—i.e., "You have rights, and they are as follows." However, what the Office of the Ombudsperson has recommended is a more extensive service where patients can receive rights *advice* from an independent third party who is not involved in their treatment. The rights advisor serves only the patient, who may confidentially ask their advisor questions about their rights and their legal options for their specific situation.

### Arguments for independent rights advice in BC

Advocates for involuntary patients' rights argue that expecting clinicians to give patients rights information presents a conflict of interest: clinicians who want to keep a patient in the hospital when the patient would like to leave may, whether consciously or unconsciously, be reluctant to give the patient the information they would need to challenge their detention or treatment plan [5]. One aim of having independent rights advisors is to allow patients to receive their rights information from an unbiased third party. Proponents of independent rights advice argue that this service is necessary because clinicians are not trained in the law and may not have a good understanding of their legal obligations or of patients' legal rights [5]. As a result, they may give patients inconsistent or incorrect information, which can undermine trust in the therapeutic relationship.

In the early 1990s,

> *the Legal Services Society funded the Mental Health Law Program to provide independent rights advice to all newly admitted patients at Riverview Hospital [a psychiatric hospital in Metro Vancouver] and to the psychiatric units of several Lower Mainland general hospitals. At these locations, there were protocols in place to notify the Mental Health Law Program when patients were detained and a legal advocate came to the hospital to meet with the patients, provide them with information on their rights and take instructions from them to assist them in exercising their rights [5, p. 60].*

In 1994 the provincial Office of the Ombudsman wrote in its report that "We spoke to several former patients who said that having an independent person tell them of their rights as an involuntary patient made a huge difference to their sense of security and well being" [14, p. 4–20]. However, since the release of that report, the Legal Services Society cancelled funding for independent rights advice for involuntary patients. Currently the Forensic Psychiatric Hospital is the only facility in BC that still offers rights advice to detainees [5].

## Approaches to rights advice in other jurisdictions

Much like BC at the time of this writing, many jurisdictions—including Manitoba, Canada [15, s. 32(1)]; South Africa [16, s. 17]; Tasmania, Australia [17, s. 129]; and Ireland [18, s. 16]—rely on hospital staff and medical practitioners to inform involuntary patients about their rights and do not have independent rights advisors or patient advocates. We found a few jurisdictions that have legislated these additional rights safeguards, and their varied approaches may be instructive to BC's implementation of its rights advice service.

**Proactive (opt-out) model of rights advice.**   Several Canadian jurisdictions have independent advocacy offices that provide patients with rights advice. New Brunswick [19], Newfoundland and Labrador [20], Nova Scotia [21], Ontario [22], and Saskatchewan [23] have rights advisors who must be independent of the treatment team and, in some cases, independent of the health authority. These rights advisors automatically meet with involuntary patients and explain their rights to them. In Saskatchewan, rights advisors are legal professionals, whereas in New Brunswick, they tend to be social workers.

A 2005 evaluation of rights advice services from Ontario's Psychiatric Patient Advocate Office for people on community treatment orders found general satisfaction with independent rights advisors among clinicians and patients, though not among all family members:

> *Many CTO [Community Treatment Order] clients and substitute decision-makers told us that they appreciated the rights advice that they received. With the exception of a minority of family members who oppose all forms of legal protection for patients, everyone we spoke with told us that rights advice is a necessary and positive part of the process. [24, p. 98]*

A 2012 evaluation of Newfoundland and Labrador's *Mental Health Care and Treatment Act* found that most involuntary patients recall meeting with a rights advisor and said that the rights advisor helped them understand their rights. Some stakeholders in the mental health system believed, however, that the legislation's requirement that rights advisors meet with patients within 24 hours of admission meant that rights advisors saw patients when they were not in a state of mind to understand their rights information:

> *In general, all participants across disciplines were in support of the rights advisor role and thought that it is a step forward in patient rights. The major concern from many of the participants was the timing of when the rights advisor first meets with the patient. It was thought that the rights advisor meets with the patient too soon after the patient is certified. [25, p. 37]*

In response to this evaluation, the legislation was amended so that rights advisors would give rights advice within 24 hours of admission but also follow up with the patient about their rights after 10 days [25].

Western Australia's *Mental Health Act* (2014) created the Mental Health Advocacy Service, which is required to visit patients within 7 days of involuntary admission if they are over 18 years of age and within 24 hours if they are under 18 years of age to ensure they understand their rights and to help them exercise their rights. However, the legislation also requires

hospital staff to give patients information about their rights when they are first involuntarily admitted [26].

According to a 2018 review of the implementation of the *Mental Health Act* (2014):

*Ten out of 24 (42%) consumer responses advised that they did not have their rights explained to them [by hospital staff] in a form, language and terms they understood. [27, p. 22]*

It is unclear from the review why so many patients report not being explained their rights. On one hand, hospital staff might not be as diligent in giving rights information if they know that advocates will eventually contact every involuntary patient—although in that case a possible wait of 7 days seems long relative to other jurisdictions. On the other hand, the results of this review seem to underscore the necessity of an independent advocacy service as a safeguard to ensure that patients do get information about their rights.

Beyond advocating on behalf of individual patients, Western Australia's Mental Health Advocacy Service also offers education to clinicians about the necessity of giving involuntary patients rights information:

*[T]he MHAS provides presentations to mental health staff regarding the importance of explaining rights to consumers. However, the MHAS acknowledges the frequency of these presentations could be increased. [27, p. 22]*

**Reactive (opt-in) model of rights advice.** In Alberta, Canada, involuntary patients can request independent rights advice from the Mental Health Patient Advocate. In 2019 a court ruled that this provision of the *Mental Health Act* was unconstitutional, a violation of the *Canadian Charter of Rights and Freedoms*' section 7 right to life, liberty, and security of the person [28]. Justice Kristine Eidsvik wrote that rights advice should be given to patients proactively, without their having to request it. The province appealed this ruling [29], but the Alberta Court of Appeal upheld it [30], a decision that may have implications for Canadian jurisdictions that don't currently offer proactive rights advice.

In England and Wales, the 2007 amendments to the *Mental Health Act* established Independent Mental Health Advocates (IMHAs), whose role is to help patients understand and exercise their rights. Hospital staff and medical practitioners are required by the legislation to inform patients about the advocacy service "as soon as practicable after the patient becomes liable to be detained." [31].

An independent review of the IMHA service found that the patients who would most benefit from advocacy—namely, racialized populations, people with dementia, people with learning disabilities, patients who are deaf or hard of hearing, and children and youth—were least likely to access the service. As a remedy for these inequities, the reviewers recommended moving from an opt-in system, where patients must request advocacy services, to a proactive, opt-out system:

*We interviewed several qualifying patients who were unable to recall seeing any information about the IMHA service or of being given any information by staff. Some mental health professionals were also vague about what information was provided and when and what they needed to know. [32, p. 72]*

*Consideration should be given to establishing an opt-out rather than an opt-in system to promote access to IMHA services and ensure that those most in need of IMHA services are not missing out on the opportunity. This should involve consultation with service user groups,*

*particularly with those with experience of compulsion, carers and organisations that represent specific interest—BME [Black and minority ethnic] communities, children and young people, older people, people with learning difficulties, people with physical disabilities or sensory impairments and LGBT [lesbian, gay, bisexual, and trans] people. [32, p. 248]*

Similarly, an evaluation of the Independent Mental Health Advocacy service in Victoria, Australia, recommended a shift to a proactive, opt-out system for greater accessibility, noting that such a shift would require additional investment:

*To successfully maintain the rights of people subject to compulsory mental health treatment IMHA must be accessible to everyone who is eligible for it. This requires the adoption of an opt-out system where every person made subject to compulsory treatment is offered advocacy. This will require increased resourcing for IMHA to be able to respond to increased demand. [33, p. v]*

## Relevance to the present study

As this overview shows, the approaches to ensuring that involuntary patients receive their rights information vary enormously within Canada and internationally. Many jurisdictions, as BC does currently, assign the responsibility to clinicians only. Some jurisdictions, such as Alberta, Canada, and England and Wales, UK, have independent patient advocates who can help patients understand and exercise their rights, but only reactively—that is, upon the patients' request. Finally, a small number of jurisdictions, including Ontario and Nova Scotia, Canada, and Western Australia, offer involuntary patients proactive rights advice, where independent advisors visit every involuntary patient to ensure that they have been informed of their rights.

The Ombudsperson's recommendation for BC to adopt an independent rights advice service represents a major shift for the province, introducing new people and a new process to involuntary admissions. Because of its possible effects on clinical workflow and patient care, we sought to understand clinicians' perspectives on this proposed independent rights advice service.

## Methods

Between 2018-05-24 (YYYY-MM-DD) and 2018-07-27, clinicians who give rights information at three hospitals in Vancouver, BC, were invited to participate in one of five sessions featuring focus groups and training about the *Mental Health Act*. Clinical nurse educators or clinical nurse specialists in the psychiatric units at those hospitals promoted the sessions for the month leading up to them, via posters in high-traffic staff areas and via email. In these session promotions and at the sessions themselves, participants were told that this study was being done as a part of the first author's doctoral research.

Over a three-hour open house, the training session rotated between three 15-minute segments: (a) a focus group, (b) a didactic training session about clinicians' responsibilities to give rights information under the *Mental Health Act*, and (c) a question-and-answer period about the *Mental Health Act*. As a result, each of these segments was offered four times over the course of the open house. Clinicians could join the session at the start of any 15-minute segment and experience the full curriculum by remaining for 45 consecutive minutes. Participants attended these catered sessions over their lunch breaks.

For the focus group component of the session, participants were asked to reflect on their role in giving rights information to involuntary patients, and they were specifically asked for their opinions on independent rights advisors as follows: "How would you feel about a model where a rights advisor who was independent of the hospital visited involuntary patients after admission and gave them information and advice about their *Mental Health Act* rights?" We chose to use focus groups for data collection because the clinicians were already congregating for training, and adding a group discussion would allow us to efficiently probe their opinions and attitudes on a topic based on their professional experience [34]. We anticipated that interaction among a group of colleagues would generate rich discussion about a topic that has been studied relatively little. The focus groups also allowed the facilitator to build rapport with the participants, helping to increase their receptivity to the training curriculum [35].

The first author—a female non-clinician doctoral candidate with no lived experience of involuntary hospitalization but with training in qualitative research and three years' interviewing experience—facilitated the discussion, while an undergraduate- or master's-level research assistant took notes. The focus group discussions were audio recorded, and the first author transcribed the recordings verbatim, using the research assistant's notes to check the accuracy of the transcript and to annotate the transcript with participants' nonverbal communication. Shortly after each session, the first author analyzed the transcript using thematic analysis [36]. Thematic analysis "is a method for identifying, analysing and reporting patterns (themes) within data" [36, p. 79] and can be used inductively or deductively, while accommodating a variety of theoretical frameworks. In this study, the transcripts were analyzed inductively to identify themes related to independent rights advice, informed by an interpretive description framework [37]. The first author coded the transcripts using NVivo 12 [38], following Braun and Clarke's [36] 15-point checklist for good thematic analysis. The aim was to hold enough focus group sessions to achieve thematic saturation.

This study received ethics approval by the University of British Columbia Behavioural Research Ethics Board (UBC BREB number H18-00132). Participants provided written consent.

## Results

Eighty-one clinicians—including nurses, social workers, occupational therapists, and other clinical professionals—participated in the focus groups over the five open house sessions. The mean number of participants in the 15-minute focus groups was 5.1, with the smallest group having 2 participants and the largest 9 participants. The sessions were not mandatory, so the sample was self-selected. Other than the clinical nurse educators and clinical nurse specialists who had coordinated with the first author to advertise the sessions in their respective units, the participants had no prior relationship with the first author. Table 1 shows the number of participants at each session based on profession. The 8 participants on 2018-07-17 were clinicians working at a rapid-assessment unit, whereas the other sessions were attended by clinicians from acute and tertiary care units. Table 2 summarizes the participants' self-reported years in clinical practice. Psychiatrists and other physicians were invited, but none participated.

Each participant was assigned a unique number identifier. The first two sets of digits refer to the month and day of the session, and the last set of digits was incremented with every participant who joined the session. (For example, 07-27-12 participated on July 27 and joined after participant 07-27-11.) This identifier is used in the supporting quotes that follow.

Because the facilitator and participants had established a rapport through the training segments of the open house sessions, and because the participants were coworkers at the same facility who self-selected to participate as a result of their interest in the topic, the focus group

**Table 1. Number and profession of focus group participants at each session.**

| Session date | Number of participants in each profession | | | | Total participants |
|---|---|---|---|---|---|
| | Nursing | Social work | Occupational therapy | Other profession* | |
| 2018-05-24 | 13 | 1 | 0 | 4 | 18 |
| 2018-05-29 | 13 | 1 | 2 | 10 | 26 |
| 2018-06-12 | 11 | 1 | 2 | 1 | 15 |
| 2018-07-17 | 5 | 3 | 0 | 0 | 8 |
| 2018-07-27 | 12 | 2 | 0 | 0 | 14 |
| **TOTAL** | **54** | **8** | **4** | **15** | **81** |

* Other self-reported professions included clinical counsellor, dietician, nursing student, occupational therapy student, pharmacist, psychiatric worker, psychologist, recreational therapist, rehabilitation worker, research assistant, and site coordinator.

discussions flowed smoothly and featured considerable interaction among participants [35]. Thematic saturation was reached after four sessions—the data from the fifth session produced no new themes. The dataset of anonymized focus group transcripts is available via Simon Fraser University's data repository at https://researchdata.sfu.ca/pydio_public/43fd77.

The themes we identified that related to independent rights advisors are as follows.

## Clinicians' scope of practice

Many of the clinicians believe that giving rights information is within their scope of practice, although some would appreciate regular opportunities to refresh their training.

*07-27-12 (nurse)*: *I think as nurses, like, we're more than capable to actually provide the rights, but, like [07-27-10, a nurse] said, maybe just doing, like, maybe an in-service once a month or once every few months so newer nurses can go in and kind of go in and sit on that and have that information accessible to them, would be of value. [. . .] It is within our scope of practice, but, yeah, just to have more information maybe behind it if you're not familiar with it.*

\*\*\*

*07-17-02 (nurse)*: *I think also that our social workers and nurses, given the right education, like this kind of thing, would have the tools to provide that information. I think we have the capability to provide that. At least at that level. [. . .] It's within our scope, is what I'm saying. Yeah.*

Some clinicians even saw the interaction as an opportunity to show that they are acting in a patient's best interests so that they can cultivate a therapeutic relationship.

**Table 2. Focus group participants' self-reported number of years in clinical practice.**

| Session date | Participants with number of years in clinical practice | | | | |
|---|---|---|---|---|---|
| | < 1 | 1–5 | 6–10 | > 10 | Unreported |
| 2018-05-24 | 3 | 6 | 3 | 6 | 0 |
| 2018-05-29 | 4 | 9 | 2 | 9 | 2 |
| 2018-06-12 | 3 | 3 | 3 | 6 | 0 |
| 2018-07-17 | 1 | 4 | 1 | 2 | 0 |
| 2018-07-27 | 0 | 6 | 6 | 2 | 0 |
| **TOTAL** | **11** | **28** | **15** | **25** | **2** |

*05-24-00 (nurse)*: *I think there's also a lot of benefits to nurses giving it [rights information] because it can really help you show that you're on their side—like, "Not only am I on your side; this is how you exercise your rights."*

## Conflict of interest

But several participants also acknowledged the conflict of interest between their role as clinicians and as staff members of an institution detaining involuntary patients. Some participants suggested that pressure to give patients incomplete information sometimes comes from the physicians who admit the patients and who don't want the administrative burden of having their decisions challenged by the patient at a review panel hearing.

*05-29-19 (psychologist)*: *It's kind of like a dual relationship, or, you know, like, you have a conflict of interest, right? You're the clinician, but then you're also trying to enforce the law. Do you know what I mean? Anyways. It's kind of a dual role.*

***

*06-12-12 (nurse)*: *When you're offering something like "you have the right to a review panel." Physicians don't want review panels. They take up a lot of time. They take up a lot of work. So they're going to. . . you know, in the information that the patient's getting, maybe they're glazing over that part about review panel because they don't want to have to spend the X amount, you know. So to me that is a direct conflict of interest, and having a rights advisor who can come in and openly give you all that information without any sort of motivation or bias, that would be the ideal system, I think.*

## Clinicians' concerns about independent rights advisors

The participants' chief concerns about independent rights advisors relate to:

- when patients can receive rights information,

- how additional people would be accommodated in the clinical workflow, and

- whether demand for review panel hearings would increase unnecessarily.

Some clinicians worried that patients would have to wait too long to hear about their rights if rights advisors were not immediately available.

*07-27-04 (nurse)*: *Realistically, an advocate isn't going to be coming in on a long weekend, so the person could sit there for several days without their rights, if we were counting on that. [. . .] Yeah, I think, I mean, like, an expert's great, but if we defer to some expert, that's just going to delay people getting their rights. They should be getting their rights as soon as possible.*

***

*06-12-05 (nurse)*: *And sometimes it can be affected by length of stay, like, if you have a short-term patient waiting around for those rights, they may be discharged by the time someone shows up, which would not be helpful.*

But other clinicians believed that having independent rights advisors would not necessarily result in a delay in patients' receiving rights information, especially if the advisors were on site:

*07-27-08 (nurse)*: *I think this idea that timely rights information and a third party don't need to be mutually exclusive. Like, I think we could potentially have someone with that level of expertise in house, like, given the number of involuntary patients we have. So I think that's just a consideration that we often exclude from the conversation.*

For clinicians in a rapid-assessment unit, in particular, where patients are assessed, admitted, then immediately transferred to another psychiatric unit, waiting for an independent rights advisor would not be feasible.

*07-17-05 (social worker)*: *I don't think it would work in our setting.*

*07-17-02 (nurse)*: *Yeah, because. . . it's just. . . there's no planning involved in terms of when somebody walks in and needs to be certified, so it would be. . . somebody that we'd have to have, like, on call, almost, to come in at random times.*

Several participants thought that having to integrate an independent rights advisor, particularly someone external to the hospital, into the admission process would lead to too much confusion in an already chaotic situation. The rights advisor might not be familiar with how the psychiatric unit functions, accountability to ensure patient understanding may be hard to record and track, and rights advisors may give patients information that is inconsistent with hospital-level policies and patient privileges.

*07-27-05 (nurse)*: *We have a problem to just communicate with ourselves, then you have to add another person in there and then have a conversation with them about, "Did they understand? What did you tell them? What questions did you ask?" And where's that documented —are they putting that in the chart? Or you're gonna go and put that in the chart? I don't know. It seems like a big added thing.*

\*\*\*

*07-17-08 (nurse)*: *I think, yeah, it would be great to, like, totally have that [an independent rights advisor] as a resource, but I'd be curious about how it would of play out adding another person to that process, where it's just kind of, like, "Who are these people? What is happening?"*

\*\*\*

*05-24-00 (nurse)*:. *If they [independent rights advisors] ever gave a message that was inconsistent, then it's a potential for creating conflict. [. . .] We're the ones who permit or don't allow certain things within the hospital, so I could see there being an area of conflict. If they're, like. . . Nothing in the Mental Health Act says that you have to be in your PJs [pajamas]. Yet, our policy here as you come in, and you're first in PJs, you know, and it's not a question at the beginning, right? And it's, you know, a pass, a privilege or whatever it is, but it doesn't say that explicitly in the Mental Health Act, so when you have inconsistencies that way, that's the only part where it could be a disadvantage.*

One clinician recalled the period when some BC hospitals did have independent rights advisors and said that their advocacy led to some patients applying for review panel hearings early on in their hospitalization when they didn't truly understand the function of the hearings, only to cancel them when their mental state improved.

*06-12-02 (social worker)*: *I don't know the numbers in terms of how many people that are patients might actually follow through with hearing their rights, requesting a review panel, and going through a review panel. I don't know what. . . how the numbers go over that. But, um, with those people [independent rights advisors] coming, it was always higher at the beginning, because they were advocating as well as presenting. And so it was always more paperwork and more forms and then more people saying, "Oh, I don't want to do that" as they started to. . . as they started to feel better and kind of fully understand that they wouldn't be here for a month. [. . .] So that was a concern with that system.*

## Clinicians' reasons to support independent rights advice

Despite some clinicians' misgivings, most participants recognized positive aspects to independent rights advisors. Many participants felt that they lacked the legal training themselves to understand the complexities of patients' rights and answer their legal questions, and they said that having a third party to take on the task of giving patients rights advice would allow clinicians to focus on building a therapeutic rapport with patients.

*07-27-09 (nurse)*: *I think that in order to provide rights to patients accurately, you have to be well versed in those rights and the implications of them. [. . .] I would feel more comfortable if I, one, was able to actually get training that would support that depth of knowledge if I was going to be the one providing rights, or two, I would defer to somebody who had that expertise whether it would be a third party or it be someone on the team. [. . .] The default shouldn't be just us [nurses, giving rights information] because we're afraid of a third party being on the unit or somebody else. Like, we need to default to who has the best information and who's current in all that information. I don't think I'd be able to speak to sort of, like, the human rights law associated with involuntary certification and ramifications of that. I don't have the legal training. Just like we wouldn't expect a lawyer to come in and do. . . complete a nursing pass that they've never done or had training on.*

\*\*\*

*06-12-03 (nurse)*: *I love the idea of it [independent rights advice], because, again, it takes that even just like stigma from first-line responder to patients about talking about rights. . . takes that off of us and places that on to someone like a lawyer or a paralegal so we can have more of an intimate kind of therapeutic relationship with our clients—that we're not focusing about stripping their rights, we're focusing about giving them rights. So I love that. I can't believe that we don't have it anymore, 'cause that just seems like a no-brainer, kind of. So whether it's, like, a single shift a week that somebody's here or just maybe management has kind of been trained in that regard that they can kind of come down to the ward or there could be a group or something. . . something about that that's just readily available could be beneficial helping with providing rights.*

Most participants saw independent rights advisors playing a role in offering patients (a) an option of hearing rights information from someone outside of their treatment team if they wished or (b) an escalating level of service, where clinicians could be the first to give rights information but more in-depth rights advice would be available upon the patients' request.

*05-29-12 (nurse)*: *I think it's good to give patients an option. Do you want external people to explain to you or what we explain to you is good enough? I think if it's an option for the patients and the family, I think that would be good. Yeah.*

\*\*\*

*05-29-00 (nurse)*: *I also think sometimes having an outside person come in, like, if they're really paranoid about the hospital or all of us kind of being in cahoots together, for some people that might be more therapeutic. Sometimes I think it would be more therapeutic if it came from somebody they had rapport with.*

\*\*\*

*07-27-10 (nurse)*: *I think my idea would be [. . .] would almost be, like, a continuum, like, we give, like, frontline staff and nurses, giving those rights, with the caveat then saying at the end of these rights, saying, "If you need more information, we could set up an appointment with this legal, uh, professional or legal expert, so that you could, like, better understand it," or have, like, a. . . almost like a set thing, like, so that person checks in with, like, every person who potentially could use more information. It's almost like multilevel, but, like, on a continuum. So, like, it's available to everyone who wants it.*

Some participants believed that an independent office could be a valuable resource for clinicians, either offering regular training about the *Mental Health Act* or answering rights questions that arise from interactions with patients.

*07-17-06 (social worker)*: *I mean, I think it'd be amazing to have that resource as clinicians. Like, when you have. . . because I remember, like, reading through the Mental Health Act and thinking, like, "I'm not a lawyer." [. . .] So it would be nice, like, with some of the complexities of it, if somebody. . . if the client is asking you a question and you just don't know, to be able to consult.*

\*\*\*

*07-27-14 (social worker)*: *I think a third party being involved would also potentially be a useful tool for helping people understand their rights. Or also maybe providing in-services to staff to kind of jog their memory about what specifically those rights are and how to articulate those.*

## Discussion

Most participants supported the idea of some type of role or organization, independent of the treatment team, that could give patients rights advice or serve as a resource for clinicians who wanted *Mental Health Act* training or had legal questions about patients' rights. Some participants—in the rapid-assessment setting especially—felt that an independent rights-advice service would not be compatible with their clinical workflow. Clinicians' concerns about how an independent rights-advice service would be implemented in BC could be informed by other jurisdictions' approaches, including:

- distinguishing between basic rights information and rights advice,
- establishing reasonable timelines for rights advice, and
- using on-site versus external rights advisors.

### Distinguishing between basic rights information and rights advice

Opponents to independent rights advice have argued that section 10 of the *Charter* requires only that patients be *informed* of their rights, with no further obligation on the facility's part to

facilitate patients' exercising their rights. They believe that clinicians are best suited to give patients this rights information because they are always available and are trained to interact with people in acute mental distress [12]. Further, they argue that giving this information presents no more of a conflict of interest than a police officer telling a person they arrest about their rights [13].

However, the Office of the Ombudsperson's recommendation was for rights *advice* from an independent third party, a service through which an involuntary patient could not only get information about their involuntary status and their rights but also receive confidential counsel about the legal options they have available based on their specific situation. Making the clear distinction between this rights advice and clinicians' obligations to give basic rights information, which they could continue to offer, would address some clinicians' concerns.

All Canadian jurisdictions that offer proactive independent rights advice—namely, New Brunswick, Newfoundland and Labrador, Nova Scotia, Ontario, and Saskatchewan—also explicitly require in their mental health legislation that patients receive rights *information* from staff upon detention at the hospital. This basic information, which fulfills section 10 of the *Charter*, includes where the patient is, why they are being detained, and that they have the right to a lawyer. Rights advisors then meet with the patient—"promptly" (Ontario [22] and Saskatchewan [23]), "as soon as possible" (Nova Scotia [21]), within 24 hours (Newfoundland and Labrador [20]), or within 72 hours (New Brunswick [19]) after admission.

Currently, in BC, clinicians are expected to give patients this information but also answer rights questions and help them exercise their right to request to a review panel hearing or second medical opinion [1]. Although clinicians can provide more than basic rights information, they cannot offer confidential counsel or advocacy that a rights advisor with deeper knowledge of the legislation would be able to provide.

We recommend establishing a clear distinction between giving rights information and giving rights advice in BC. Promptly upon a patient's involuntary admission, regardless of the time or day, clinicians can ensure that the patient is promptly given information about their rights and the reason for their detention. Independent rights advice can then be offered later.

This distinction would also help address clinicians' concerns about clarifying clinicians' versus rights advisors' roles, responsibilities, and accountability tracking in delivering rights information to patients.

## Establishing reasonable timelines for rights advice

As for when patients should receive rights advice, Newfoundland and Labrador's evaluation of its service suggested that many patients are too unwell to understand or remember in-depth rights advice for the first 24 hours after admission [25]. Although some former involuntary patients in BC reported wanting to be notified of their upon admission, others acknowledged that they were too agitated or overwhelmed at the start of their hospitalization to understand their rights, with some describing the process as frightening because they believed being told about their rights meant they were in legal trouble. Many of these patients described the importance of having multiple opportunities to learn about their rights throughout their hospitalization [11].

Clinicians in the focus group also worried that telling people about their right to challenge their certification before they are fully able to understand their situation could lead to a higher volume of review panel applications, Many of these applications would later need to be cancelled, increasing administrative burden on staff.

BC's *Mental Health Act* is structured in a way that could reasonably accommodate a delay between admission and rights advice [4]. Patients are detained under the authority of a Form 4

medical certificate. One certificate allows a patient to be detained for up to 48 hours. If, within that time, a second certificate is completed, the patient may be detained for up to one month. Certain *Mental Health Act* rights, including the right to apply for a review panel hearing and the right to request a second medical opinion, are not available to patients until the second certificate is completed.

According to the legislation, patients must receive rights *information* after both the first and the second certificates. But waiting until the second certificate to give patients independent rights *advice* would allow patients up to 48 hours to become aware of their situation in hospital, and it would allow the patients to hear about their rights at the time those rights become fully available to them. In this scenario, clinicians in rapid-assessment units would likely not have to accommodate an independent rights advisor at all, focusing instead on giving only basic rights information. The inpatient psychiatric units receiving the involuntary patients from the rapid-assessment units would then coordinate rights advice.

In practice, first and second certificates are often completed at the same time, so a mechanism such as the one in Newfoundland and Labrador [20], where rights advisors make a follow-up visit 10 days after admission, would give patients another opportunity to learn about their rights when their mental state may have improved.

## Using on-site versus external rights advisors

Clinicians expressed concern that rights advisors could cause confusion in clinical workflow, especially if they are external to the hospital and are not well versed in the facility's policies. This problem could be mitigated with on-site rights advisors. These advisors would be familiar with the hospital's admissions process and work alongside the treatment team to give patients rights advice at an appropriate time, with consideration to their mental state.

The recommendation to use on-site rights advisors is supported by Ontario's experiences in implementing rights advice through its Psychiatric Patient Advocate Office:

> *It is recommended that the advocate be located "in-house" rather than externally but must remain independent of the facility. This means that advocates should have their offices in the facility and operate as much as possible from this location. This will keep advocates in close contact with the facility and allow them to become more familiar with facility policies and regulations. It also may provide increased opportunities for rapport building between advocates and facility staff [39].*

On-site rights advisors may not be justifiable in terms of cost for facilities with low volumes of involuntary hospitalization. At those sites, rights advice offered via telephone or videoconferencing may be an option.

## Limitations

A few factors may have biased the data collected. First, the focus groups were voluntary, and the topic was well advertised in advance. It's possible that clinicians who are already strong advocates for involuntary patients' rights self-selected to participate, whereas clinicians who believe in restricting patients' rights may have avoided the sessions.

Second, although all clinicians who give rights information were invited to participate, no physicians chose to participate in the study. In BC, physicians are the only clinicians authorized under the *Mental Health Act* to complete the medical certificates to detain involuntary patients. Physicians may not have chosen to participate because (1) the sessions were promoted by nursing staff, and physicians may not have realized the relevance of the topic to their

practice; (2) the sessions were not accredited to offer physicians Continuing Medical Education credits for their participation, which would count toward their credential maintenance requirements; and (3) they lacked time to participate, a reason they cite as the top barrier to engaging with continuing professional development or quality improvement activities [40]. Dedicated efforts to reach this key group of stakeholders may yield further insights that could affect implementation of independent rights advice. That said, giving involuntary patients rights information has in practice overwhelmingly been done by social workers and nurses [41], and it is the nursing workflow and day-to-day patient care that would likely be most affected by the introduction of an independent rights advice service.

Further, although the focus group data are anonymized, participants engaged in a discussion with their coworkers at the focus groups, and unseen power dynamics may have prevented participants from expressing themselves frankly. Supplementing focus groups with one-on-one interviews may have mitigated the effects of these power dynamics.

The 15-minute focus group segments may have limited in-depth discussion and richer data might have been gathered in longer sessions. However, the fact that we achieved thematic saturation after four 15-minute open house sessions suggests that our method successfully identified the key issues related to the discussion topic. In their evaluation of the open house session [35], participants who suggested possible improvements to the format wanted more time for the question-and-answer segment and not the focus group segment of the program. Only one respondent specifically asked for "More time for discussion," and it is not definitive which of the two segments they were referring to.

The focus groups involved clinicians at a small number of hospitals within a single metropolitan area. Clinicians in other parts of the province, particularly in rural and remote locations, may have quite different opinions about the value and feasibility of an independent rights-advice service.

Finally, this analysis does not account for cost, a chief concern of the opponents of an independent rights-advice service. A province-wide economic analysis considering various implementation options, including on-site staff and remote rights advice, may complement these stakeholder consultation findings.

## Conclusions

BC clinicians who participated in focus groups about their roles in giving rights information to involuntary patients were generally amenable to the idea of independent rights advisors, with some expressing optimism that these advisors would be a valuable resource for clinicians to increase their own knowledge of the legislation and to give their patients better care. However, some participants did raise concerns about how rights advisors would affect their clinical workflow.

Many of those concerns could be addressed by distinguishing between rights *information*, which should continue to be given by staff to patients upon admission, and rights *advice*, to be given by a preferably on-site but independent rights advisor after the second medical certificate is completed.

This study demonstrates the insights that can be gained by consulting with stakeholders who, along with patients, will be among the most affected by an independent rights-advice service. Addressing their key concerns as the service is still being conceived may directly address perceived barriers and increase clinician buy-in so that implementation is more likely to succeed. Similarly engaging other stakeholders, including family care partners and a broader range of clinicians, including physicians, will offer policy makers further insights as they develop this service.

## Acknowledgments

Thanks are due to focus group research assistants Emily Carpenter, Alyssa Haim, Alicia Lee, Hannah Rosen, and Jennifer Tong. Thanks also to Vanessa Bland, Amy Byrne, Philip Charlebois, and Angela Russolillo for facilitating site access and advertising the sessions to clinicians.

## Author Contributions

**Conceptualization:** Iva W. Cheung, Diego S. Silva, Kimberly J. Miller.

**Data curation:** Iva W. Cheung.

**Formal analysis:** Iva W. Cheung.

**Funding acquisition:** Iva W. Cheung, Kimberly J. Miller, Erin E. Michalak.

**Investigation:** Iva W. Cheung.

**Methodology:** Iva W. Cheung, Diego S. Silva, Kimberly J. Miller, Erin E. Michalak, Charles H. Goldsmith.

**Project administration:** Iva W. Cheung, Erin E. Michalak.

**Supervision:** Diego S. Silva, Kimberly J. Miller, Erin E. Michalak, Charles H. Goldsmith.

**Validation:** Iva W. Cheung.

**Writing – original draft:** Iva W. Cheung.

**Writing – review & editing:** Iva W. Cheung, Diego S. Silva, Kimberly J. Miller, Erin E. Michalak, Charles H. Goldsmith.

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
