## [Decision Letter · Decision Letter 0]

21 Oct 2020

PONE-D-20-26216

A qualitative study of clinicians’ perspectives on independent rights advice for involuntary psychiatric patients in British Columbia, Canada

PLOS ONE

Dear Dr. Cheung,

Thank you for submitting your manuscript to PLOS ONE. After careful consideration, we feel that it has merit but does not fully meet PLOS ONE’s publication criteria as it currently stands. Therefore, we invite you to submit a revised version of the manuscript that addresses the points raised during the review process.

We look forward to receiving your revised manuscript.

Kind regards,

Nikola Biller-Andorno

Academic Editor

PLOS ONE

Additional Editor Comments:

Dear authors,

Thank you for a nicely written paper. It is a pleasure to see state-of-the-art qualitative work submitted to PLOS ONE. As you will see from the reviewers' comments there are a few issues that should be addressed to strengthen the paper further.

Best regards,

Nikola Biller-Andorno

Journal Requirements:

Reviewers' comments:

Reviewer's Responses to Questions

**Comments to the Author**

1. Is the manuscript technically sound, and do the data support the conclusions?

Reviewer #1: Yes

Reviewer #2: Yes

Reviewer #3: Yes

2. Has the statistical analysis been performed appropriately and rigorously? 

Reviewer #1: N/A

Reviewer #2: N/A

Reviewer #3: N/A

3. Have the authors made all data underlying the findings in their manuscript fully available?

Reviewer #1: Yes

Reviewer #2: Yes

Reviewer #3: Yes

4. Is the manuscript presented in an intelligible fashion and written in standard English?

Reviewer #1: Yes

Reviewer #2: Yes

Reviewer #3: Yes

5. Review Comments to the Author

Reviewer #1: Dear authors,

congratulations on this interesting manuscript! My main point of criticism is the sole focus on Canada, limiting its relevance for readers from other countries. Please find additional comments below:

Major points:

Consider adding information on the frequency of certification in BC to the introduction to give the reader (who may not be from Canada) a sense of the scope of the issue.

To me, the paragraph in lines 58-61 would be better suited for the background section, as it is an argument pro independent rights advice.

To me, the structure of your background section is somewhat confusing – I recommend sorting it into pro and con arguments. Also, a discussion of ethical arguments is lacking (principle of respect for autonomy).

In order to make your manuscript more relevant to readers from countries other than Canada, I recommend reviewing (or citing a review of) the situation in other jurisdictions, too (briefly, and possibly limited to the US and Europe, f.ex.)

In your methods section, please justify the choice of focus groups for your project.

As you repeatedly refer to participants working in a rapid-assessment setting in your results section, I believe you should report on how many participants worked in such a setting in your methods section.

Did you collect data on whether the participants of your study had work experience in a setting with legal rights advice service (f. ex. In another jurisdiction or in the context of the project you describe from line 126 on). As such experience would likely influence participants’ views, I believe it would be important information for your sample description.

In my experience, 15 minutes is a rather short time frame for a focus group. Please report the average number of participants per individual focus group including a measure of variance and comment on whether participants felt they had enough time to respond to the introductory question (“How would you feel about a model where a rights advisor…”).

I believe that the lack of physicians in your sample is a serious limitation to your study and should be discussed. Another limitation is the lack of service user participation.

Minor points:

Lines 178-179: “In 2019 a court ruled that this provision of the Mental Health Act unconstitutional [20].“ I believe this sentence is missing a verb (but I am not a native speaker of English, so if you are, feel free to ignore this comment).

Reviewer #2: Thank you for giving me the opportunity to review this manuscript, which I find clearly written, methodologically sound and relevant both from a clinical as well as from a mental health policymaking perspective.

Besides some minor points (see below) there is one major issue I would like to raise: Overall, the manuscript seems to be conceived from a very practical, hands-on point of view on the topic. Neither the background nor the discussion include some more theoretical reflections on the issues at stake. As it stands, the results are very much in line with this i.e. the themes being raised by the participants are primarily practical in nature. There is however one practical concern, namely the timing of providing legal information and advice, which entails an important ethical challenge on further reflection, I believe: Admitting a person against her will is only justified if this person is in severe need of treatment (and, at least according to some legislations, lacks capacity to consent to it). But arguably, understanding - and even more so acting on - legal information and advice requires similar cognitive and emotional abilities than the capacity to decide for or against treatment. Of course, capacity is always situational and decision-/topic-specific and of course a person who lacks capacity in some respects has rights about which she has a right to know. But - and here comes the ethical challenge - whilst promoting patients' rights and autonomy might seem ethically imperative, might it not also be unethical because it overburdens someone with decisions that she is incapable of taking? Even though the participants did not raise this fundamental challenge, I think the theme of 'timing' and participants' (yet shallow) reflections on how to inform and advise patients in a way that they can understand invites a more in-depth consideration of this issue in the discussion.

Minor points:

- I would add 'involuntary admission' or something alike to the keywords.

- Please provide some detail about the legislation on detainment in the background section. Who can detain a person in BC (psychiatrists, any medical doctor...? one person alone or is there always a second opinion)? A the doctors who detain the person/who issue the certificate working on the unit(s) on which the person is then treated or is there a separation between detaining and treating physician and institution? For how long is a person detained for? What are the legal requirements for detention (is lack of capacity one of them)? I realise that some of this information is offered later on in the manuscript, but it is an important background for the reader against which to consider the results and it helps to have all of this information in the beginning rather than being offered it bit by bit.

- Overall, the design seems well-suited to assess the issue in question. However, 15 minutes are extremely short for any kind of in-depth discussion. Please comment on this and provide some description of the flow/the dynamics of the conversation in the focus group sessions.

- Please discuss the absence of psychiatrists and other physicians in the sample in more detail. As you note yourself, this a major limitation of the study which should be reflected thoroughly.

Reviewer #3: The submitted research article for potential publication in the PLOS ONE “A qualitative study of clinicians’ perspectives on independent rights advice for involuntary psychiatric patients in British Columbia, Canada” addresses a very important question from a clinical, legal, and ethical point of view. The aim of the study was to assess within a qualitative study design how clinicians feel about the recommendation of an independent body (instead of clinicians) to give rights advice to patients upon admission.

The aim of the study is well-defined. However, the research questions could be stated more precisely (see additional comment below).

The methods are appropriate and well described.

The data are sound. Although I am not an expert of the qualitative methods used, I am easy able to follow the presentation and interpretation of the findings.

The manuscript adheres to the relevant standards for reporting and data deposition. However, the statement that “The dataset of anonymized focus group transcripts is available via Simon Fraser University’s data repository at https://researchdata.sfu.ca/pydio_public/43fd77.” should be stated in the manuscript.

The discussion and conclusions are balanced and adequately supported by the data.

Limitations of the work are clearly stated (see additional comment below).

The title and abstract do accurately convey what has been studied.

The writing is good.

Suggested Minor Revisions:

1) The study focuses on the situation and law in BC, Canada. It makes sense that the authors provide the reader with the legal background on rights advice elsewhere in Canada. However, for the readership of this international journal, it would be interesting if the issues were embedded in the context of the international conceptual literature on involuntary hospitalization and rights advice. In addition, a short overview on the legal background in other constitutional democracies could be provided (e.g., the US, European countries, Australia, NZ etc.)

2) It would help the reader if the research question(s) were stated more precisely. The sentence on the lines 80-82 (“This paper presents focus group findings on the perspectives of clinicians who currently give patients rights information about the possibility of independent rights advisors.”) seems rather a summary for the abstract, and the applied methods belong to the Methods section. What follows the above-mentioned sentence on the lines 80-82 is the Background section. It would make more sense to state the research question(s) based on the Introduction AND the Background sections just before the Methods section.

3) Even for a qualitative study, the results section is very long with more than 3,000 words (more than 17,000 letters). Consequently, the authors should shorten this section.

- In the Limitations section, the authors state that “Second, although all clinicians who give rights information were invited to participate, the study was not able to engage any physicians, and they are the only clinicians authorized under the Mental Health Act to complete the medical certificates to detain involuntary patients.” In fact, no psychiatrists, no other physicians, and only one psychologist participated in the focus group. I consider this THE major limitation of the study, and the authors should discuss the reasons for that fact.

6. PLOS authors have the option to publish the peer review history of their article (what does this mean?). If published, this will include your full peer review and any attached files.

Reviewer #1: **Yes: **Anna Lisa Westermair

Reviewer #2: **Yes: **Anke Maatz

Reviewer #3: **Yes: **Manuel Trachsel, MD, PhD

---

## [Author Response · Author response to Decision Letter 0]

11 Jan 2021

Reviewer #1, Anna Lisa Westermair

1. Dear authors,

congratulations on this interesting manuscript! My main point of criticism is the sole focus on Canada, limiting its relevance for readers from other countries. Please find additional comments below: 

Thank you! Our response to the critique about our focus on Canada is below, comment 5.

2. Consider adding information on the frequency of certification in BC to the introduction to give the reader (who may not be from Canada) a sense of the scope of the issue. 

We have added certification figures for the 2016–2017 fiscal year.

3. To me, the paragraph in lines 58-61 would be better suited for the background section, as it is an argument pro independent rights advice. 

Thank you for this suggestion. To us, these lines are pro rights information generally and are not necessarily relevant to pro vs. con independent rights advice. Please see also our response to your comment 4 below to understand why we have chosen to leave it where it is.

4. To me, the structure of your background section is somewhat confusing – I recommend sorting it into pro and con arguments. Also, a discussion of ethical arguments is lacking (principle of respect for autonomy). 

Thank you for this suggestion.

We have experimented with a few rearrangements of the background section and ultimately concluded that the section on practices in other jurisdictions is not so much a list of pro vs. con arguments. Rather, it highlights various approaches that can be taken in a framework of independent rights advice, each with its own advantages and disadvantages.

However, we do take the point that the first section on arguments for and against independent rights advice in BC could be more coherent.

As a result, we have rearranged our background section to feature the following sections:

• arguments against independent rights advice in BC

• arguments for independent rights advice in BC

• approaches to rights advice in other jurisdictions (with examples of reactive and proactive models)

• relevance to the present study 

We hope the clear delineations between sections and added headings will be less confusing than the original draft.

As for a discussion of ethical arguments, we agree that involuntary hospitalization and rights information raise numerous important ethical issues. We have added allusions to the ethical aspects of independent rights advice but feel that an in-depth ethical discussion has been explored in other work and is beyond the scope of this study.

5. In order to make your manuscript more relevant to readers from countries other than Canada, I recommend reviewing (or citing a review of) the situation in other jurisdictions, too (briefly, and possibly limited to the US and Europe, f.ex.) 

We have added some information about rights-advice practices from the UK and Australia, which have similar approaches to mental health legislation as much of Canada. We chose not to include the US, which relies on more of a judicial approach to involuntary hospitalization that does not compare as easily to British Columbia.

Interestingly, the findings from this exercise of looking at other jurisdictions seem to suggest that proactive, independent rights advice is not particularly common.

6. In your methods section, please justify the choice of focus groups for your project. 

We have added a justification for focus groups in our methods section.

7. As you repeatedly refer to participants working in a rapid-assessment setting in your results section, I believe you should report on how many participants worked in such a setting in your methods section. 

We have clarified in the results section that one of the sessions was attended exclusively by 15 clinicians from a rapid-assessment unit. 

8. Did you collect data on whether the participants of your study had work experience in a setting with legal rights advice service (f. ex. In another jurisdiction or in the context of the project you describe from line 126 on). As such experience would likely influence participants’ views, I believe it would be important information for your sample description. 

We did not explicitly collect these data. We believe that prior experience with independent rights advice would inform clinicians’ views rather than bias the data. Also, it is our judgment that such prior experience would not alter the findings of this study or the 

policy recommendations. 

9. In my experience, 15 minutes is a rather short time frame for a focus group. Please report the average number of participants per individual focus group including a measure of variance and comment on whether participants felt they had enough time to respond to the introductory question (“How would you feel about a model where a rights advisor…”). 

We have added the mean number of participants, as well as the minimum and maximum number of participants in our focus group sessions, which we believe may be more illuminating than variance in this case.

We have added a comment to the limitations section about the 15-minute format. In our view, achieving thematic saturation suggests that we successfully identified the key issues related to the discussion topic. Further, participants who completed session evaluations did not identify the length of the focus group as being too short. 

10. I believe that the lack of physicians in your sample is a serious limitation to your study and should be discussed. 

We have elaborated on this limitation in the text, suggesting possible reasons for a lack of physician engagement. It is an area ripe for further research.

However, because giving patients rights information and providing day-to-day care is in practice overwhelmingly a nursing responsibility, we feel that nurses’ workflow would be most affected by the implementation of an independent rights advice service. 

11. Another limitation is the lack of service user participation. 

The presence of service users in the focus groups likely would have influenced the participants’ candour. In our introduction, we write, “Patients’ perspectives of the rights-information process have been presented elsewhere,” along with a reference to the publication where those findings are described.

12. Minor points:

Lines 178-179: “In 2019 a court ruled that this provision of the Mental Health Act unconstitutional [20].“ I believe this sentence is missing a verb (but I am not a native speaker of English, so if you are, feel free to ignore this comment). 

Thank you. We have added a verb to the sentence.

Reviewer #2, Anke Maatz

1. Thank you for giving me the opportunity to review this manuscript, which I find clearly written, methodologically sound and relevant both from a clinical as well as from a mental health policymaking perspective. 

Besides some minor points (see below) there is one major issue I would like to raise: Overall, the manuscript seems to be conceived from a very practical, hands-on point of view on the topic. Neither the background nor the discussion include some more theoretical reflections on the issues at stake. As it stands, the results are very much in line with this i.e. the themes being raised by the participants are primarily practical in nature. There is however one practical concern, namely the timing of providing legal information and advice, which entails an important ethical challenge on further reflection, I believe: Admitting a person against her will is only justified if this person is in severe need of treatment (and, at least according to some legislations, lacks capacity to consent to it). But arguably, understanding - and even more so acting on - legal information and advice requires similar cognitive and emotional abilities than the capacity to decide for or against treatment. Of course, capacity is always situational and decision-/topic-specific and of course a person who lacks capacity in some respects has rights about which she has a right to know. But - and here comes the ethical challenge - whilst promoting patients' rights and autonomy might seem ethically imperative, might it not also be unethical because it overburdens someone with decisions that she is incapable of taking? Even though the participants did not raise this fundamental challenge, I think the theme of 'timing' and participants' (yet shallow) reflections on how to inform and advise patients in a way that they can understand invites a more in-depth consideration of this issue in the discussion. 

This is an interesting point of discussion, but it is complicated in many ways by BC’s somewhat unusual legislation.

BC is the only province in Canada whose Mental Health Act does not include a test for competency. Thus, someone can be perfectly lucid but still meet the criteria for involuntary hospitalization. In our previous study involving interviews with former patients, some participants who were at risk of self-harm or suicide met the criteria for certification but were completely aware of their surroundings. These participants generally preferred to get rights information immediately upon admission. A delay of giving rights information to accommodate patients who may not be ready for it would disadvantage this population.

We feel that an ethical discussion of this topic would be interesting but somewhat beyond the scope of this particular study. We have added (a) allusions to the ethical aspects of independent rights advice in the introduction, (b) a clarification to the introduction that explains that the law requires staff to attempt giving patients information again at a later time if they appear not to be able to understand their rights when they are first admitted, and (c) a short description of patients’ views of the timing of rights information from another study of ours that supports your concern that patients may feel overwhelmed at the time of admission.

That said, we feel that our recommendations of assigning staff the duty of giving simple rights information (i.e., “This is where you are, and this is what it means to be certified.”) and independent rights advisers arriving up to 48 hours after admission to give rights advice (i.e., “This is how you can legally challenge your detention.”), as well as allowing for more than one visit by the rights advisor, adequately addresses this concern about overburdening patients with decision they are not in the right state of mind to make.

2. Minor points:

- I would add 'involuntary admission' or something alike to the keywords. 

Added. Thank you for the suggestion.

3. - Please provide some detail about the legislation on detainment in the background section. Who can detain a person in BC (psychiatrists, any medical doctor...? one person alone or is there always a second opinion)? A the doctors who detain the person/who issue the certificate working on the unit(s) on which the person is then treated or is there a separation between detaining and treating physician and institution? For how long is a person detained for? What are the legal requirements for detention (is lack of capacity one of them)? I realise that some of this information is offered later on in the manuscript, but it is an important background for the reader against which to consider the results and it helps to have all of this information in the beginning rather than being offered it bit by bit. 

We have added information about the criteria for involuntary hospitalization to the introduction.

4. - Overall, the design seems well-suited to assess the issue in question. However, 15 minutes are extremely short for any kind of in-depth discussion. Please comment on this and provide some description of the flow/the dynamics of the conversation in the focus group sessions. 

We have addressed this point in our response to Reviewer #1, comment 9.

5. - Please discuss the absence of psychiatrists and other physicians in the sample in more detail. As you note yourself, this a major limitation of the study which should be reflected thoroughly. 

We have elaborated on this limitation. Please see our response to Reviewer #1, comment 10 for further details. 

Reviewer #3, Manuel Trachsel, MD, PhD

1. The submitted research article for potential publication in the PLOS ONE “A qualitative study of clinicians’ perspectives on independent rights advice for involuntary psychiatric patients in British Columbia, Canada” addresses a very important question from a clinical, legal, and ethical point of view. The aim of the study was to assess within a qualitative study design how clinicians feel about the recommendation of an independent body (instead of clinicians) to give rights advice to patients upon admission. 

(No response needed.)

2. The aim of the study is well-defined. However, the research questions could be stated more precisely (see additional comment below). 

Addressed below in response to comment 11.

3. The methods are appropriate and well described. 

(No response needed.)

4. The data are sound. Although I am not an expert of the qualitative methods used, I am easy able to follow the presentation and interpretation of the findings. 

Always good to hear!

5. The manuscript adheres to the relevant standards for reporting and data deposition. However, the statement that “The dataset of anonymized focus group transcripts is available via Simon Fraser University’s data repository at https://researchdata.sfu.ca/pydio_public/43fd77.” should be stated in the manuscript. 

We have added this statement to the results section.

6. The discussion and conclusions are balanced and adequately supported by the data. 

(No response needed.)

7. Limitations of the work are clearly stated (see additional comment below). 

(No response needed.)

8. The title and abstract do accurately convey what has been studied. 

(No response needed.)

9. The writing is good. 

(No response needed.)

10. Suggested Minor Revisions:

1) The study focuses on the situation and law in BC, Canada. It makes sense that the authors provide the reader with the legal background on rights advice elsewhere in Canada. However, for the readership of this international journal, it would be interesting if the issues were embedded in the context of the international conceptual literature on involuntary hospitalization and rights advice. In addition, a short overview on the legal background in other constitutional democracies could be provided (e.g., the US, European countries, Australia, NZ etc.) 

We have added context from other countries, including South Africa, Australia, and the UK. Approaches to involuntary hospitalization vary widely among nations—and even within Canada. For example, some provinces allow involuntary patients to refuse treatment, whereas British Columbia does not. We feel that a discussion of legal differences among Canadian and international jurisdictions is beyond the scope of this study.

We have omitted the US from our discussion because its approach to involuntary hospitalization is more judicial in nature to Commonwealth countries and not as easily comparable.

11. 2) It would help the reader if the research question(s) were stated more precisely. The sentence on the lines 80-82 (“This paper presents focus group findings on the perspectives of clinicians who currently give patients rights information about the possibility of independent rights advisors.”) seems rather a summary for the abstract, and the applied methods belong to the Methods section. What follows the above-mentioned sentence on the lines 80-82 is the Background section. It would make more sense to state the research question(s) based on the Introduction AND the Background sections just before the Methods section. Thank you for this suggestion. 

We have made our research question more explicit but kept it in the introduction, because we feel it is important to orient the reader to the thrust of the study early in the article.

We reiterated the research question and study objectives immediately before the methods, as suggested.

12. 3) Even for a qualitative study, the results section is very long with more than 3,000 words (more than 17,000 letters). Consequently, the authors should shorten this section. 

We have shortened or deleted many of the illustrative quotes. What remains we feel is important for context.

13. - In the Limitations section, the authors state that “Second, although all clinicians who give rights information were invited to participate, the study was not able to engage any physicians, and they are the only clinicians authorized under the Mental Health Act to complete the medical certificates to detain involuntary patients.” In fact, no psychiatrists, no other physicians, and only one psychologist participated in the focus group. I consider this THE major limitation of the study, and the authors should discuss the reasons for that fact. 

We have elaborated on this limitation. Please see our response to Reviewer #1, comment 10 for further details.

---

## [Editor Report · Decision Letter 1]

4 Feb 2021

A qualitative study of clinicians’ perspectives on independent rights advice for involuntary psychiatric patients in British Columbia, Canada

PONE-D-20-26216R1

Dear Dr. Cheung,

We’re pleased to inform you that your manuscript has been judged scientifically suitable for publication and will be formally accepted for publication once it meets all outstanding technical requirements.

Kind regards,

Nikola Biller-Andorno

Academic Editor

PLOS ONE
---

## [Editor Report · Acceptance letter]

10 Mar 2021

PONE-D-20-26216R1 

A qualitative study of clinicians’ perspectives on independent rights advice for involuntary psychiatric patients in British Columbia, Canada 

Dear Dr. Cheung:

I'm pleased to inform you that your manuscript has been deemed suitable for publication in PLOS ONE. Congratulations! Your manuscript is now with our production department. 

Kind regards, 

on behalf of

Professor Nikola Biller-Andorno 

Academic Editor

PLOS ONE